# Review of Constructions and Materials Used in Swedish Residential Buildings during the Post-War Peak of Production

**Björn Berggren * and Maria Wall**

Department of Architecture and Built Environment, Division of Energy and Building Design, Lund University, Box 118, 221 00 Lund, Sweden; maria.wall@ebd.lth.se

*   Correspondence: bjorn.berggren@ebd.lth.se

**Abstract:** One of the greatest challenges for the world today is the reduction of greenhouse gas emissions. As buildings contribute to almost a quarter of the greenhouse gas emissions worldwide, reducing the energy use of the existing building stock is an important measure for climate change mitigation. In order to increase the renovation pace, there is a need for a comprehensive technical documentation that describes different types of buildings in the existing building stock. The purpose of this study is to analyse and describe existing residential buildings in Sweden. The data are based on published reports from 1967 to 1994 that have not been publicly available in a database for other researchers to study until now. Data from the reports have been transferred to a database and analysed to create a reference for buildings and/or a description of building typology in Sweden. This study found that there is a rather large homogeneity in the existing residential building stock. However, it is not possible to use a single reference building or building technique to cover the majority of the existing buildings. In Sweden, common constructions for exterior walls in multi-dwelling buildings which should be used for further studies are insulated wood infill walls with clay brick façades, lightweight concrete walls with rendered façades and concrete sandwich walls. The most common constructions for one- and two-dwelling buildings are insulated wooden walls with clay brick façades or wooden façades. Furthermore, roof constructions with insulated tie beam and roof constructions where the tie beam is a part of the interior floor slab are frequently used and should be included in further studies.

**Keywords:** renovation; residential buildings; reference building; building stock data base

---

## 1. Introduction

One of the greatest challenges for the world today is the reduction of greenhouse gas emissions. Worldwide, energy use in buildings accounts for over 40% of the world's primary energy usage and 24% of the greenhouse gas emissions [1]. Within the European Union (EU), Switzerland and Norway, the largest portion of the housing stock is residential houses, with a current growth rate of around 1% [2]. Thus, even if policy-makers set strict energy requirements for the new construction of residential houses, the effect may be low. Hence, reducing the energy use in the existing building stock is an important measure for climate change mitigation. Specifically, in Sweden, there are 4.5 million homes and the Swedish National Board of Housing, Building and Planning (Boverket), estimated that 75 percent of these dwellings must undergo major renovations before 2050 [3]. Boverket further concluded that the pace of renovation of existing buildings must increase as an important climate change mitigation measure.

Several projects and studies have demonstrated that there is a large potential to reduce the energy demand in existing buildings by improving the building envelopes and technical installations [4–23]. A

recent study from Italy also described different approaches for U-value assessment, including infrared thermography, which may be used to assess the U-value of constructions in existing buildings [24]. Detailed analyses for energy performance can be found in different studies [9,12–16,20–23] where energy performance was investigated for cities, boroughs and buildings. However, the description/input data collected from the buildings are generic in most of the mentioned studies. Descriptions of building envelopes are presented as U-/R-values, rather than describing the construction. Only four of the mentioned studies present the building envelopes and/or renovation measures in more detail [20–23].

Studies have also shown that many of the existing buildings in Europe were built between 1940 and 1980, and opportunities to use prefabricated building systems for the energy renovation of these building envelopes have been identified [17–19,25]. These opportunities are partly based on the finding that there is a large homogeneity in this building segment (buildings constructed between 1940 and 1980) [4,18,19,26,27]. In Sweden, this homogeneity is partly because during the most intense construction period, the so-called "Miljonprogrammet" (the million homes programme), the undertaking of projects with 1000 or more apartments in similar buildings was encouraged through rules on mortgages [26]. The million homes programme represents the era that was driven by the 1966 Swedish Parliament's decision that one million new homes would be built in a decade (1965–1974) [28].

However, there is a risk that the assumption of homogeneity is too general, since it is largely based on the architectural design of the buildings, i.e., not looking into differences among different regions within Sweden and differences regarding load bearing structure. For example, it may be possible to renovate a lightweight concrete exterior wall with a rendered façade by applying additional insulation on the exterior side of the wall followed by a new layer of rendered façade. The same measure cannot be applied on a lightweight infill wall with a ventilated façade. A more regional analysis is important in the Swedish context, since many contractors focused on renovation may be active in a limited region.

Another issue concerning building envelope improvement is that even if an increased thermal resistance of building envelopes can improve the energy performance of existing buildings, it is seldom profitable [4,9], especially concerning the exterior walls. Thus, there is a need for the development of more cost-effective methods or prefabricated building elements that can substantially increase the thermal resistance of the building envelopes of existing buildings. To enable this development, there is a need for a comprehensive technical document that describes different types of buildings in the existing building stock in Sweden as well as the building systems and materials used.

In Sweden, such a technical description exists and includes the method of production and materials used, among other aspects. Between 1967 and 1994, Statistics Sweden (SCB) [29] published a compilation of the data every year, covering from 1962 up until 1992 [30]. However, the yearly-published data are not available in a database, which makes the information difficult to analyse in a generic way for renovation purposes. The data come from applications for state loans, where technical descriptions of the buildings were made based on a predefined template. The loans, granted by the state, ended in 1992 [28].

It should be noted that the idea of describing a building stock or reference buildings is not new. The recast of the energy Performance of Buildings Directive (EPBD) [31] required that member states set minimum requirements for energy performance based on optimal cost levels, highlighting the need for reference buildings to allow optimal cost levels to be defined. An extensive research project, TABULA [32], was conducted in 2009–2012. It included the creation of residential building typologies for 13 European countries in order to make the energy refurbishment processes in the European housing sector transparent and effective. Within this project, a report describing the Swedish situation was published. The report defined three different geometries for residential buildings. Based on these three building geometries, the effects of different energy-saving measures were investigated [33]. In many of these cases, the construction of the existing exterior walls was not defined.

Two general Swedish reference buildings were defined by The Swedish National Board of Housing, Building and Planning (Boverket) in 2010 [34]. Later, based on the EPBD recast, Boverket defined ten reference buildings in order to calculate optimal cost levels for energy performance in 2013 [35].

However, only five different geometries and three different exterior wall constructions were defined, as other parameters varied, such as ventilation and heating systems.

A Special Issue of the journal *Energy and Buildings* was recently published with a focus on monitoring, mapping and modelling the existing building stock in Europe [36–46]. However, of the previous studies that compiled and described a specific building typology, only one Spanish case [46] where the constructions of different buildings were described more in detail was found.

There are several Swedish-based studies that describe the building typology and renovation measures more in detail [20,21,47–49]. However, some concerns still exist. Three of the studies [20,47,48] based their topologies on previous work carried out by Boverket [34], who produced an inventory of 1400 residential buildings. However, it was pointed out that it is unclear whether these buildings are statistically representative or not [20]. Furthermore, none of these three studies present differences in building typology based on geographic location. The fourth study [21] uses reference houses without any specification regarding underlying data for these. The last of the mentioned studies [49] bases its building typology on architectural books [50,51], which describe architectural trends rather than statistical data.

As mentioned at the beginning of the introduction, Boverket estimated that 75 percent of the existing homes in Sweden must undergo major renovation before 2050 [3]. There are no specific data related to how existing homes are renovated today. However, the inventory carried out by Boverket in existing buildings [34] shows that the amount of insulation in existing buildings is low compared to Swedish regulations today. Buildings before 1990 can be expected to have less than 100 mm of insulation in ground constructions, less than 200 mm of insulation in exterior walls and less than 300 mm of insulation in roof constructions.

No Swedish-based study has conducted a bottom-up analysis to create reference buildings and/or a description of building typologies, including type of buildings, load bearing constructions, materials used, etc., for different regions in Sweden. A recent Swedish-based study presents a method for creating a synthetic building stock [52]. The study underlines the lack of data available for building stock characterization, which necessitates the use of a synthetic model.

Therefore, in this article, data from the SCB reports [30] were compiled to enable generic bottom-up analyses, which describe the data in order to create references for further studies. This may enable strategic development of more cost-effective and robust methods or prefabricated building elements that can substantially increase the thermal resistance of building envelopes in existing buildings. All of the compiled data is available for other researchers for further studies.

The first part of this article introduces the research problem and the purpose of the article. The second part gives an overview of definitions and nomenclature used in this study. The third part describes the available data. The fourth part describes the method used to analyse data followed by the fifth part, which presents the results. The sixth part discusses the results and compares them with previous research. In the final part of this article, conclusions and recommendations for further research are given.

## 2. Definitions and Nomenclature

Swedish residential buildings have some characteristic aspects, which were registered by SCB. These characteristics are explained below and are presented in Figures 1–3.

- Multi-dwelling building A building containing three or more dwellings. The building may be a balcony access building, point block building, slab block building or terraced building, as explained below and in Figure 1.
- One- or two-dwelling building A building containing one or two dwellings. The building may be a one-dwelling building, two-dwelling building, linked building or terraced building, as explained below and in Figure 2.
- Balcony access building A multi-dwelling building with one (or more) common staircase. The dwellings are accessed through a common balcony on each storey.

- Point block building A multi-dwelling building with one central core/common staircase in the centre of the building.
- Slab block building A multi-dwelling building with two or more common staircases.
- Terraced building A multi-dwelling building or two-dwelling building, usually with almost identical dwellings, which shares one or two walls with a neighbouring dwelling.
- One-dwelling building A building containing one dwelling.
- Two-dwelling building A building containing two dwellings, usually with almost identical dwellings stacked on top of each other.
- Linked building A number of buildings (may be more than two) which are connected via a complementary building (not used as a dwelling), such as a garage or storage area.
- Transverse load-bearing A superstructure of a building (usually slab block buildings or balcony access building) based on a system where the gable walls and interior walls are load-bearing. The load-bearing walls are oriented transversely in relation to the building's dominant longitudinal direction.
- Longitudinal load-bearing A superstructure of a building where the load-bearing walls are oriented in the same direction as the building's dominant longitudinal direction. The gable walls may also be load-bearing.
- Column construction A superstructure of a building where the dominant load-bearing wall is based on columns.
- Malmö region Includes the municipalities of Bara, Burlöv, Dalby, Genarp, Kävlinge, Lomma, Lund, Löddeköpinge, Malmö, Månstorp, Räng, Skannör, Staffanstorp, Svedal, Södra Sandby, Trelleborg, Veberöd and Vellinge.
- Göteborg region Includes the municipalities of Askim, Fjärås, Göteborg, Härryda, Kungsbacka, Kungälv, Lerum, Löftadalen, Mölndal, Nödinge, Onsala, Partille, Skepplanda, Starrkärr, Stenungsund, Styrsö, Tjörn and Öckerö.
- Stockholm region Includes the municipalities of Botkyrka, Danderyd, Djurö, Ekerö, Gustavsberg, Haninge, Huddinge, Järfälla, Lidingö, Nacka, Salem, Sigtuna, Sollentuna, Solna, Stockholm, Sundbyberg, Tyresö, Täby, Upplands-Bro, Upplands-Väsby, Vallentuna, Vaxholm, Värmdö and Österåker.
- Non-metropolitan regions Includes all municipalities except for the ones listed above.

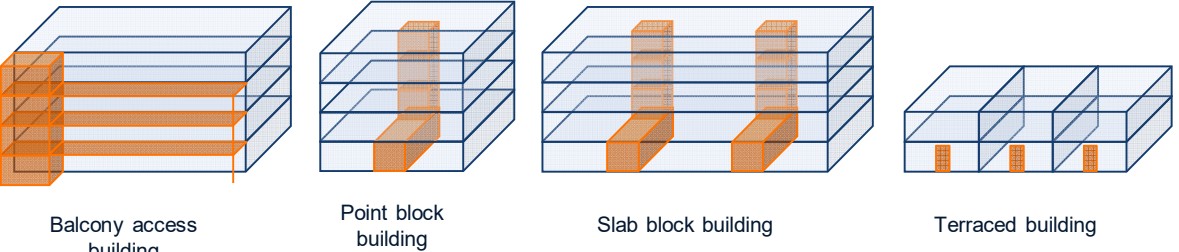

**Figure 1.** Different types of multi-dwelling buildings. The orange/dark sections represent the common areas/staircases/entrances where the residents enter their dwellings (presented in light blue).

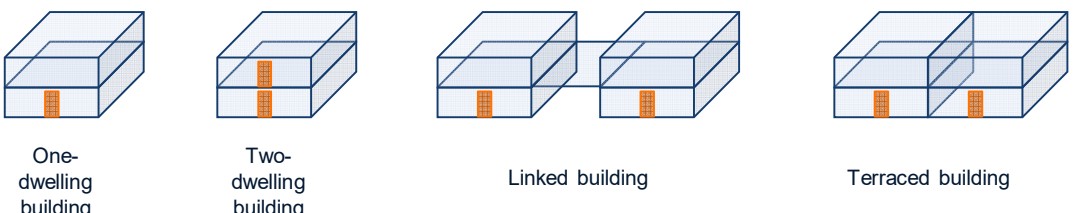

**Figure 2.** Different types of one- and two-dwelling buildings. The orange/dark areas represent entrances where the residents enter their dwellings (presented in light blue).

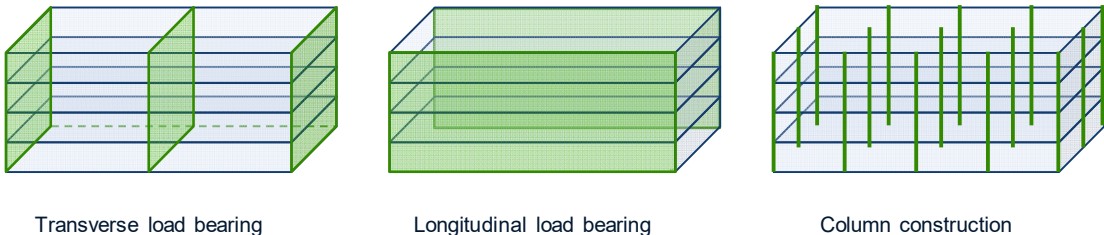

**Figure 3.** Different load bearing systems. The green/dark areas represent the load-bearing constructions.

## 3. Available Data

The available data from state loans cover the years 1962–1992 and are described in Appendices A and B.

This research focuses on residential buildings produced from 1960 up until 1990. During this period, roughly 1,250,000 dwellings were produced in multi-dwelling buildings [53]. The available data from the Statistical Reports (SR) from the state loans cover almost 1,151,000 dwellings, i.e., 92% of the produced dwellings. During the same period, roughly 900,000 dwellings were produced in one- and two-dwelling buildings. The available data cover almost 620,000 dwellings (69%). There are two main reasons for the lower coverage. The first reason is that SCB did not publish statistics from state loans for one- or two dwelling buildings in 1960–1965. The second reason is that in 1988–1990, they only published data for dwellings where the applicant of state loan was not the same as the final resident. For 1966–1987, the published data cover 83% of the dwellings. A comparison of the number of newly constructed dwellings (NC) and the data from Statistical Reports (SR) is presented in Figure 4.

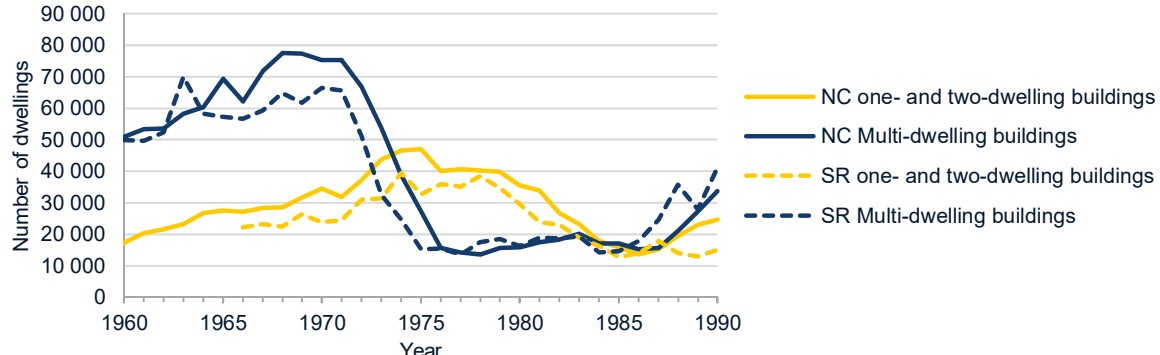

**Figure 4.** Newly constructed dwellings in Sweden. Comparison of data from newly constructed (NC) dwelling statistics [53] and Statistical Reports (SR) [27].

Figure 4 shows that the number of dwellings reported in statistical reports (SR) was sometimes higher than reported number of newly constructed dwellings (NC). This is probably due to the offset between the grant of a state loan for a building and the completion of that building.

## 4. Method

The study was carried out in four steps, as described in Figure 5. Each step is further described in this section.

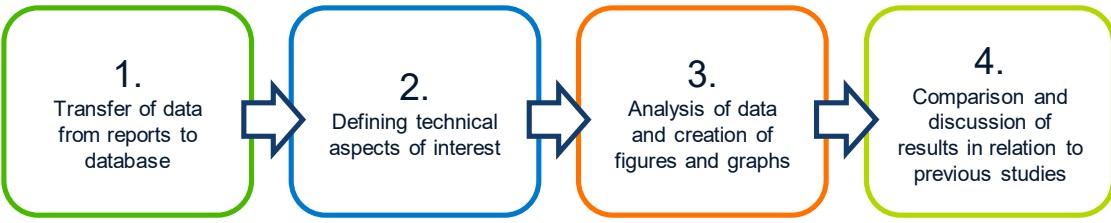

**Figure 5.** Diagram of the method used for this study.

*4.1. Step 1—Transfer of Data*

Specific data were gathered from the annual reports from SCB and transferred into a database (Excel [54]) for analysis. As data from 56 reports were transferred manually, imposing a risk of error, a quality check was carried out after the data transfer by randomly choosing ten reports and comparing the data contained in them with the data in the database.

*4.2. Step 2—Defining Technical Aspects of Interest*

Previous literature was examined to identify the most interesting technical aspects of multi-dwelling buildings and one- or two-dwelling buildings.

For multi-dwelling buildings, the type of building, number of storeys, type of superstructure and materials used for exterior walls were determined to be the most interesting technical aspects. As the roof constitutes a relatively small share of the building envelope in multi-dwelling buildings, roofs were not included in the analysis.

For one- and two-dwelling buildings, the type of building, number of storeys (including the presence of a cellar) and materials used were determined to be the most interesting technical aspects.

*4.3. Step 3—Analysis of Data*

Based on the technical aspects defined in the previous step, the existing data were analysed for different regions to create a basis for reference buildings and/or a description of building typology in Sweden. The results from step 3 are presented in the results section.

*4.4. Step 4—Comparison of Results in Relation to Previous Studies*

The results from the analysis were compared with previous research related to building typology to enable a discussion about differences between the results from this study and previous research. The comparison is presented in the discussion section.

## 5. Results

As previously mentioned, many of the existing buildings in Europe were built between 1940 and 1980. In Sweden, many of the existing dwellings were built during the so called "Miljonprogrammet" (the million homes programme). Figure 4 shows that roughly 70% of the dwellings in Sweden were built as multi-dwelling buildings during the 1960s and early 1970s. At the beginning of the 1970s, the production of dwellings in multi-dwelling buildings dropped significantly, while the production of one- and two-dwelling buildings increased, and in 1974, the production of dwellings in one- or two-dwelling buildings became higher compared to multi-dwelling buildings.

*5.1. Multi-Dwelling Buildings*

In Figure 6, the distribution of multi-dwelling buildings is presented by region and year. In 1960–1965, SCB did not present the distribution of dwellings by region. However, the total number of dwellings in Sweden was presented. To indicate the possible distribution, the data for the total amount of dwellings were based on available data for the overall distribution of dwellings in the specific decade [55]. Hence, data for that period are presented as hatched, as there is uncertainty regarding the distribution.

In 1966–1975, 59% of the dwellings in multi-dwelling buildings were produced in non-metropolitan regions. This means that they were not produced in the regions of Malmö, Göteborg or Stockholm. The production of multi-dwelling buildings dropped harshly in the mid-1970s. After a long period of low production, from the mid-1970s to the mid-1980s, an increase in the non-metropolitan regions occurred in the end of the 1980s. For the same period, no significant increase occurred in the metropolitan regions.

During the million homes programme, more than 80% of the dwellings were slab block buildings (see Figure 7). After the mid-1970s, the number of dwellings in slab block buildings remained constant

at a low level for a long period of time, with a small increase at the end of the 1980s. Instead, other building types became more common. The share of point block buildings and balcony access buildings increased, but there were also increases in other types of buildings.

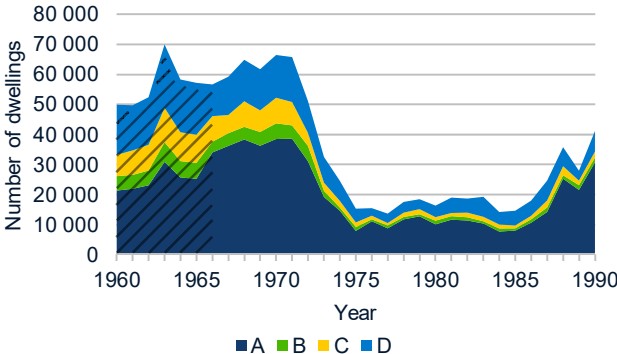

**Figure 6.** Distribution of dwellings in multi-dwelling buildings for different regions and years, as determined by state loans: (A) non-metropolitan regions, (B) Malmö region, (C) Göteborg region, (D) Stockholm region.

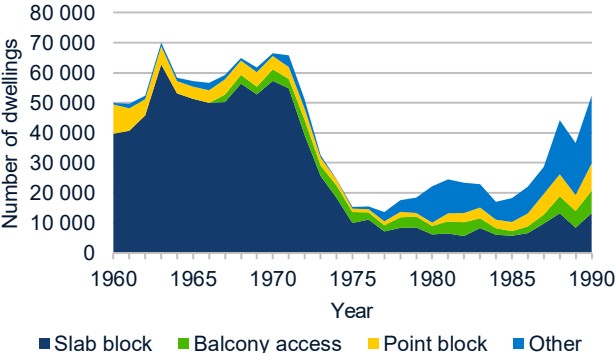

**Figure 7.** Distribution of dwellings in multi-dwelling buildings by type of building and year of state loan.

The distribution of different types of buildings in different regions was rather equal in 1966–1975 (there are no specific data for different regions for 1960–1965) with the exception of the Stockholm region, where the shares of balcony access buildings and point block buildings dwellings were higher compared to the rest of Sweden (see Figure 8). However, when the share of dwellings in slab block buildings decreased from the mid-1970s, it did not decrease as much in the Stockholm region compared to in the rest of Sweden (see Figure 8).

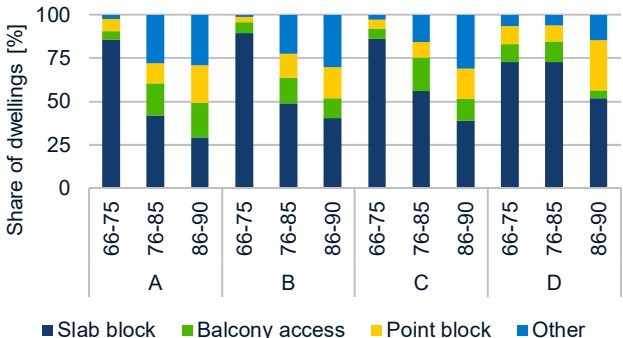

**Figure 8.** Share of dwellings in multi-dwelling buildings by type of building for different periods and regions: (A) non-metropolitan regions, (B) Malmö region, (C) Göteborg region, (D) Stockholm region.

Regarding the number of storeys, data are available for 1962–1993 (see Figure 9). Regional data for the number of storeys combined with region is available from 1968. Up until the late 1960s, dwellings in multi-dwelling buildings with three or four storeys represented more than 50% of the total dwellings. There is a clear tendency for more high-rise buildings to be built in the metropolitan regions, especially in Stockholm (see Figure 10). As much as 81% of the dwellings in multi-dwelling buildings built outside metropolitan regions are four storeys high or lower. However, in the Stockholm region, only 35% of dwellings fit into this category.

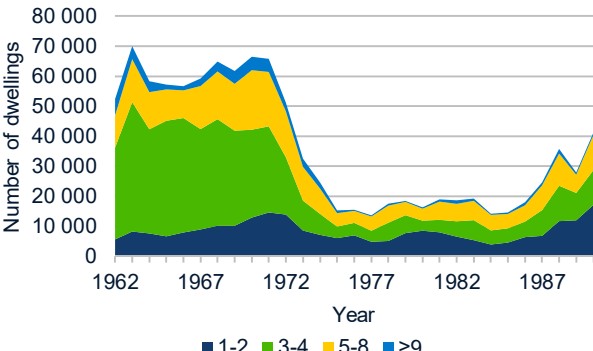

**Figure 9.** Distribution of dwellings in multi-dwelling buildings by the number of storeys and year of state loan.

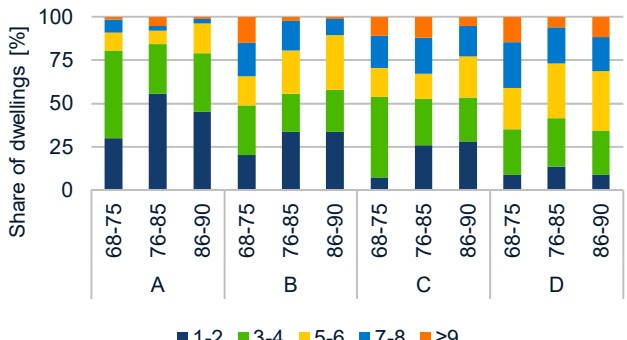

**Figure 10.** Share of dwellings in multi-dwelling buildings by the number of storeys for different periods and regions: (A) non-metropolitan regions, (B) Malmö region, (C) Göteborg region, (D) Stockholm region.

The type of superstructure was only presented by SCB for 1968–1972. However, this period is during the peak of multi-dwelling building production—the million homes programme. Therefore, it is interesting to analyse these data (see Figure 11). During this period, there was roughly a 50/50 distribution of longitudinal load bearing and transverse load bearing superstructures in the Malmö region and non-metropolitan regions. The use of transverse load bearing was roughly 10% greater in the Göteborg region and 10% lower in the Stockholm region.

In the beginning of the 1960s, residential buildings were almost exclusively designed with rendered façades or clay brick façades. The use of rendered façades reduced during the late 1960s, and concrete façades were rather frequently used during this period (see Figure 12). Throughout the analysed period, clay brick façades were most commonly used except for in dwellings with state loans from 1966 when rendered façades were used slightly more often and in dwellings with state loans from 1972 when concrete was used slightly more often.

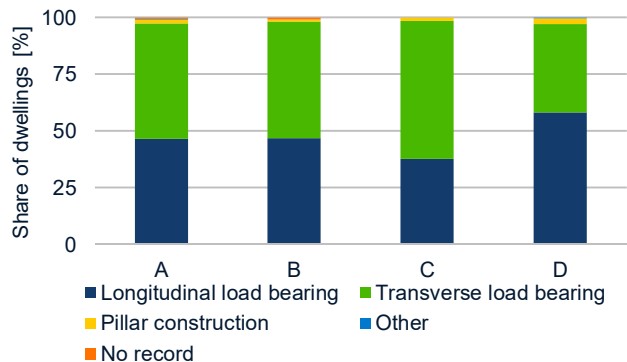

**Figure 11.** Share of dwellings in multi-dwelling buildings by type of superstructure for different regions (1968–1972): (A) non-metropolitan regions, (B) Malmö region, (C) Göteborg region, (D) Stockholm region.

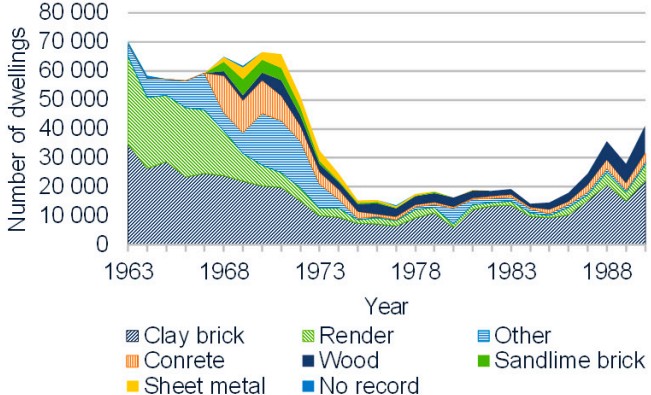

**Figure 12.** of dwellings in multi-dwelling buildings by façade material and year of state loan.

The façade materials used in different regions are presented in Figure 13. The data show that clay brick façades were not the most commonly used façade throughout Sweden for the whole analysed period. From the late 1960s to mid-1970s clay brick façades were common in non-metropolitan regions and the Malmö region, but not in the Göteborg and Stockholm regions. In the Stockholm region, rendered façades were the most common type of façade. In the Göteborg region, clay brick façades were the most common type of façade, but they were only used slightly more often than concrete façades.

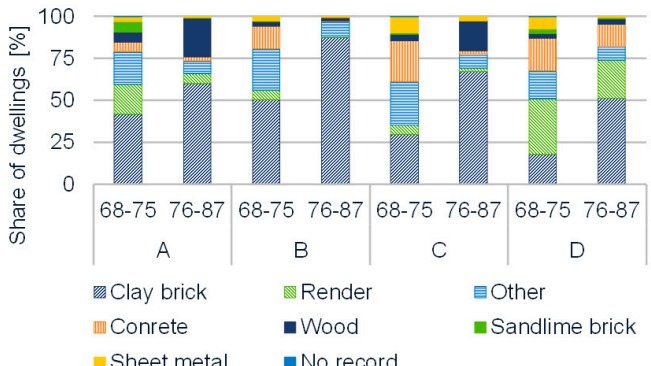

**Figure 13.** of dwellings in multi-dwelling buildings by façade material for different periods and regions: (A) non-metropolitan regions, (B) Malmö region, (C) Göteborg region, (D) Stockholm region.

In 1963–1979, SCB also published the combinations of façade material and inner material used in exterior walls, the data are shown in Figure 14. For the most common façade material, clay brick, the most common inner material was wood, followed by lightweight concrete, clay bricks and concrete.



The second most common façade material, render, was usually applied on lightweight concrete or concrete. Concrete façades were almost exclusively constructed with concrete as their inner material, except for some examples with wood and lightweight concrete. Façades of wood, sandlime brick or sheet metal were mostly designed in combination with wood as the inner material within the walls.

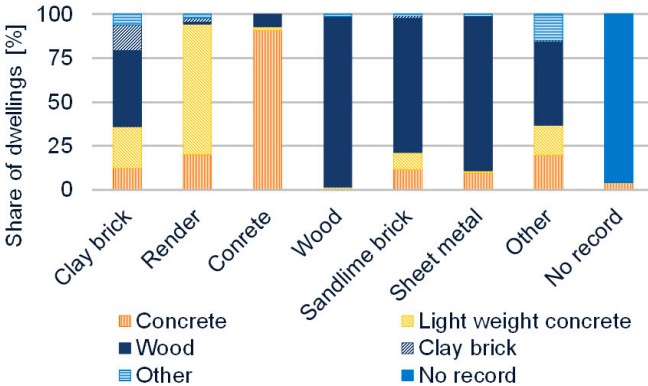

**Figure 14.** Share of dwellings by different inner material in exterior walls for different façade materials (1963–1979).

*5.2. One- and Two-Dwelling Buildings*

Regarding one- and two-dwelling buildings, the process for attaining a state loans differs depending on whether the applicant of the state loan is the final resident or not. If the applicant is not the final resident, the applicant is first given a preliminary decision before the start of the construction work. A second and final decision regarding state loans is given once the building has been completed. If the applicant is the final resident, the process is simpler, with one decision, and the applicant receives the decision about the state loan before the start of the construction work [27].

For buildings with two decisions, more data are gathered. Throughout the period where data from both one and two decisions were gathered (1966–1987), dwellings with two decisions represent 53% of the total data (see Figure 15).

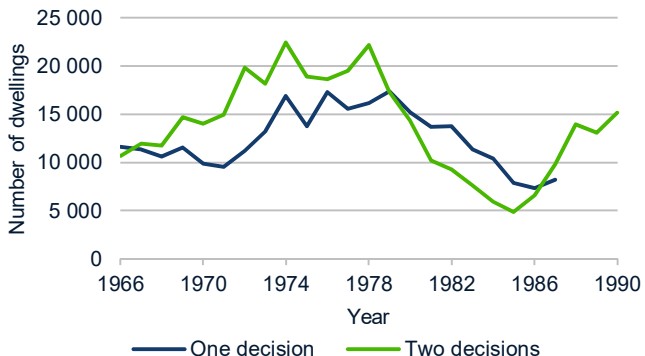

**Figure 15.** Distribution of dwellings in one- or two-dwelling buildings by one or two-decision state loans and year of state loan.

The distribution of dwellings in different regions is based on data from dwellings with two decisions (see Figure 16). In 1968–1980, 70% of the dwellings in one- or two-dwelling buildings that were given state loans following two decisions were produced in non-metropolitan regions. This means that they were not produced in the regions of Malmö, Göteborg or Stockholm. The production of one- or two-dwelling buildings dropped in the late-1970s. In the mid-1980s, an increase occurred in the non-metropolitan regions. In the metropolitan regions, no significant increase occurred.

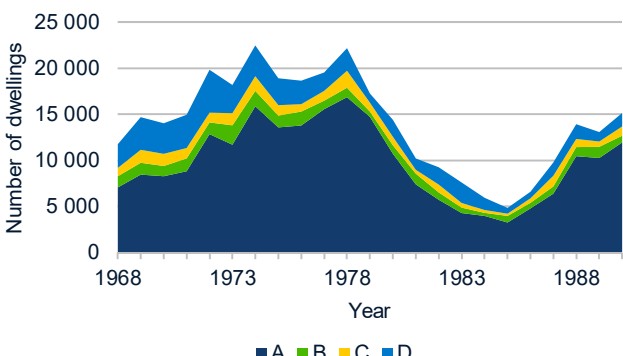

**Figure 16.** of dwellings in one- or two-dwelling buildings per region for different regions and years of state loan (with two decisions for state loans): (A) non- metropolitan regions, (B) Malmö region, (C) Göteborg region, (D) Stockholm region.

Regarding different types of one- and two-dwelling buildings, data for different types of buildings with one decision were only gathered in 1966–1967. However, 99% of the dwellings with one decision during that period were one-dwelling buildings. Based on this, it can be assumed that more than 95% of the dwellings with one decision are one-dwelling buildings.

In Figure 17, different types of buildings with two decision loans are presented together with the quantity of dwellings with one decision. One dwelling buildings together with one decision dwellings contributed to the largest share of dwellings. Together they made up between 60% and 70% of the dwellings. The largest portion of the dwellings with two decisions were terraced buildings, whose development increased significantly at the end of the 1980s. Linked buildings were rather common from the late 1960s until the mid-1970s, but their construction dropped in the late 1970s and remained rather uncommon throughout the 1980s.

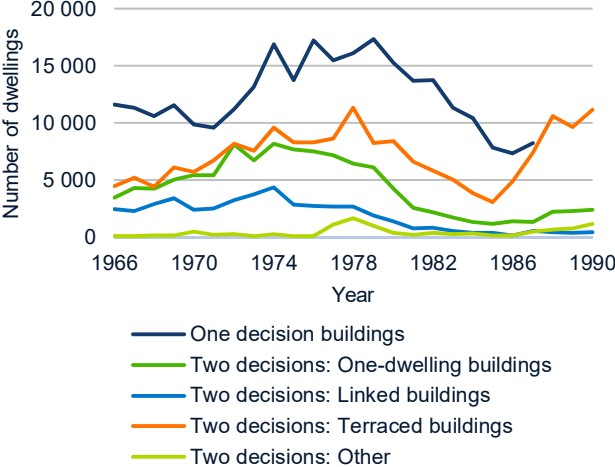

**Figure 17.** of dwellings in one- and two-dwelling buildings by type of building and year of state loan. More than 95% of the dwellings with one decision may be assumed to be one-dwelling buildings.

Compared to the Göteborg and Stockholm regions, in the Malmö region and in non-metropolitan regions, the share of dwellings built as one-dwelling buildings was rather high from the late 1960s to mid-1970s. Hence, the increase of terraced buildings had a greater effect on the distribution of different buildings in the Malmö region and in non-metropolitan regions (see Figure 18).

Regarding the number of storeys, data are available for 1970–1987 (see Figure 19). The number of storeys combined with regions is not available. At the beginning of the 1970s, dwellings in one- and two-dwelling buildings with one storey contributed to more than 60% of the total number of dwellings. However, as the production of dwellings with one storey was rather constant, the number of 1.5-storey buildings increased significantly, and in the mid-1970s, most of the state loans were given to dwellings

built with one and a half storeys. The number of dwellings built as hillside buildings and buildings with two storeys roughly varied between 2000 and 4000 dwellings/year in the 1970s. The production dropped in the 1980s and the corresponding interval was then 1000–2000 dwellings/year.

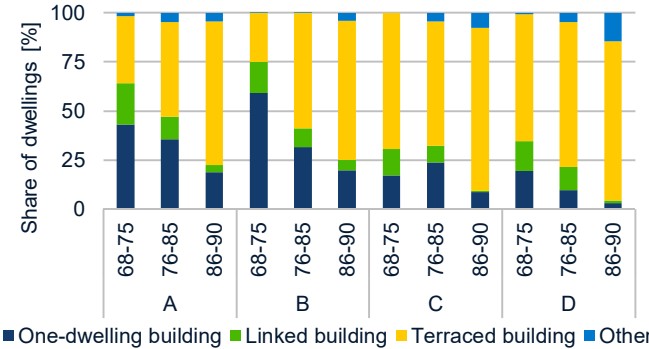

**Figure 18.** of dwellings in one- and two-dwelling buildings by type of building, for different periods and regions (with two decisions for state loans): (A) non-metropolitan regions, (B) Malmö region, (C) Göteborg region, (D) Stockholm region.

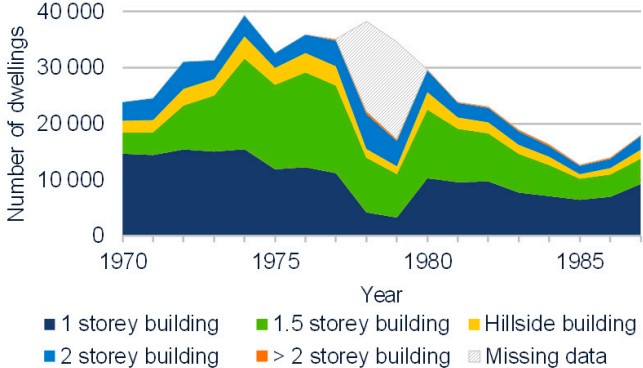

**Figure 19.** of dwellings in one- and two-dwelling buildings by number of storeys and year of state loan. Missing data refers to dwellings for which data were provided to Statistics Sweden (SCB), but the number of storeys was not specified.

Buildings with cellars were rather common at the beginning of the 1970s (Figure 20), almost 50% of the dwellings in one- and two-dwelling buildings had cellars. However, dwellings with cellars decreased during the 1970s and 1980s. In the late 1980s, almost 90% of the dwellings were built without a cellar.

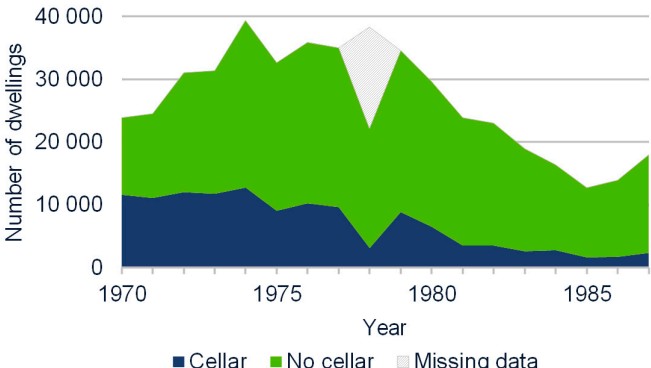

**Figure 20.** of dwellings in one- and two-dwelling buildings based on whether they had a cellar or not. Missing data refers to dwellings for which data were provided to the SCB, but information regarding cellars was not specified.

Information regarding material used for load-bearing structure in exterior walls and façade material was gathered for 1966–1987 and 1966–1990, respectively. Regarding the material used for load-bearing in exterior walls, wood was the dominant material throughout the period (see Figure 21).

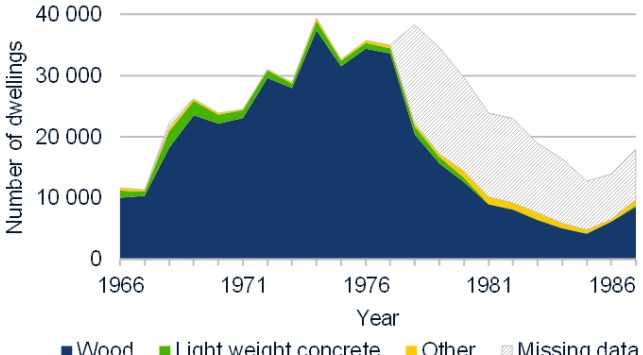

**Figure 21.** of dwellings in one- and two-dwelling buildings based on the material used for load-bearing in the exterior walls. Missing data refers to dwellings for which data were provided to the SCB, but the load-bearing material was not specified.

Regarding façade material, wood and clay brick façades were the most commonly used materials. Together, their share made up between 70% and 95% of the dwellings in 1966–1990. In the mid-1960s, façades with clay bricks were most common and accounted for almost 70% of the dwellings. The use of wood became more and more common, and at the beginning of the 1980s, wood was used for more than 70% of the dwellings, see Figure 22.

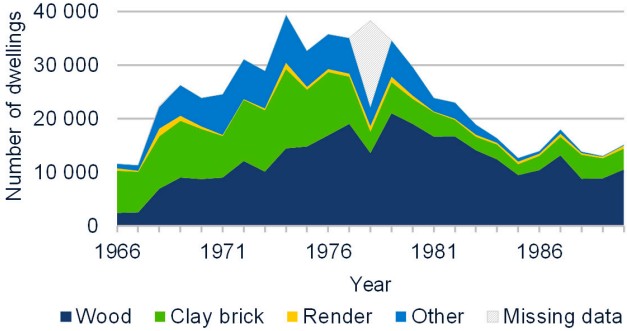

**Figure 22.** Dwellings in one- and two-dwelling buildings based on façade material. Missing data refers to dwellings for which data were provided to SCB, but the façade material was not specified.

## 6. Discussion

The data from Statistics Sweden based on Swedish state loans covers a limited period of time in Swedish history and does not cover all dwellings built during this period. However, the information is extensive and covers, to a large extent, the peak of dwelling production in Sweden, enabling a bottom-up analysis. Hence, it is interesting to compare these results with previous research related to building typology and to discuss differences. It should be noticed that previous research that involved the creation of building typologies may have had a different purpose to this research (gathering, describing and sharing data to enable further studies). For example, if the purpose of a study is to make a rough assessment of the energy performance of a building stock, not to discuss applicable refurbishment measures in detail, detailed information regarding materials used is not required.

The large share of dwellings built during the million homes programme has also been identified by previous studies as an important part of the Swedish building stock to focus on [3,11,18]. The distribution of regions corresponds rather well with previous findings [23] that 65% of the dwellings built during the million homes programme were built in non-metropolitan regions. However, after

separating all the dwellings into multi-dwelling buildings and one- and two-dwelling buildings, the data from state loans show that 59% of the multi-dwelling buildings were in non-metropolitan regions and 70% of the one- and two-dwelling buildings were built in non-metropolitan regions. It is important to highlight the rather large share of dwellings built in non-metropolitan regions, since the economic conditions are likely to be different in these regions compared to those in metropolitan regions.

*6.1. Multi-Dwelling Buildings*

The overall findings about the distribution of multi-dwelling building types (slab block, point block and balcony access buildings) correspond well to previous studies [11,18]. Furthermore, findings regarding the number of storeys also correspond rather well with previous studies [11,18,23,24]. However, it is important to highlight that even though the largest portion of the dwellings in multi-dwelling buildings from the million homes programme were to be found in slab block buildings with three or four storeys, roughly 50% of the dwellings were designed in another way. Still, many studies have based their work on a single reference building. Furthermore, it is common for multi-dwelling buildings in the metropolitan regions to have five storeys or more.

One of the most recently published studies describes 46 typical buildings of the Swedish building stock [46]. Considering multi-dwelling buildings built during the 1970s, the study defined three different buildings, all with six storeys or more. Two of these have a building footprint, which is typical of point block buildings. Looking at the available data, the point block buildings represent roughly 5% of the multi-dwelling buildings built during the 1970s. The third type of building (a slab block building with nine storeys) represents roughly 4% of the dwellings. Altogether, the defined typical buildings for multi-dwelling buildings built during the 1970s represent less than 10% of the dwellings in the building stock.

The fact that there is a rough 50/50 distribution of superstructure types is important because it provides different possibilities for energy renovation. Buildings with transverse load bearing systems can undergo major renovations to their exterior walls without major effects on the superstructure, but this is not the case for buildings with longitudinal load bearing systems. Based on statistics regarding the frequency of slab block buildings, previous studies have concluded that such buildings all use the same building technique including a transverse load bearing system and light infill walls [18,30,32]. This conclusion is wrong as the data presented here show a rough 50/50 distribution for the superstructure type.

Regarding the use of façade materials and inner material in the exterior walls, the results show that although certain materials are predominant, there is still a diverse range of materials used. For example, the most common material in walls behind clay brick façade is wood (43%). However, almost 25% of the dwellings with clay brick façades have an inner material of lightweight concrete. Clay bricks (15%) and concrete (12%) also make up for more than 25%. There are also rather large regional differences regarding common façade materials.

The TABULA study, which investigated potential energy savings in the Swedish building stock in 2012 [30], defined all existing multi-dwelling buildings as three-storey buildings and did not define the exterior wall constructions. The reason for this definition may have been an assumption that the energy-saving measures could be applied regardless of wall construction. This is a simplification that is likely not true.

In 2010, Boverket, the Swedish National Board of Housing, Building and Planning, concluded that the average multi-dwelling building in Sweden was built in 1959 [31]. This is likely to be the mean value of all multi-dwelling buildings. This is an incorrect description of the most common building in Sweden. If the analysis was based on median values instead of mean values, it would show that the most common multi-dwelling building was built during the million homes programme.

The most detailed study from Boverket [32] analysed the cost-optimal energy performance requirements for existing and new buildings. Regarding existing multi-dwelling buildings, Boverket based their analysis on two reference buildings: a three-storey building from the 1950s with lightweight

concrete exterior walls covered with render and a nine-storey building from the 1970s with concrete sandwich walls. The chosen reference exterior walls cover only approximately 25% of the existing buildings built during the million homes programme in Sweden. Furthermore, they did not include the most common exterior wall construction: wooden infill walls with clay brick façades.

It should be noted that the data used by Boverket from 1400 buildings to define their reference buildings is of high quality. The reason for the inadequate choice of reference buildings is due to poor use of the data.

## 6.2. One- and Two-Dwelling Buildings

The data for one- and two-dwelling buildings show that the production of dwellings with two decisions where the applicant was not the final resident was higher at its highest point and lower at its lowest point, compared to dwellings with one decision. This indicates that residents who build their own home are not as sensitive to the market as construction companies may be.

The fact that almost all dwellings in one- and two dwelling buildings were built with wooden constructions makes further work easier, because it means that mainly variations of façade material need to be considered in future work.

For one- and two-dwelling buildings, it will be important to study both the roof and wall constructions. Hence, it is interesting to know whether buildings are 1.5-storey buildings or not, since these buildings have very different conditions, from the perspective of adding insulation to the roof construction.

Regarding one- and two-dwelling buildings, the TABULA study based their work on two reference buildings which were both one-storey buildings with lightweight exterior walls covered with render and horizontal insulation in the roof [30]. Thus, the study does not cover any 1.5-storey buildings. This is a rather strange reference building. The chosen wall construction is likely to cover less than 5% of the existing buildings and the choice to not include 1.5-storey buildings excludes roughly 30–40% of the existing one- and two-dwelling buildings.

The study from Boverket in 2010 [31], concluded that the average one- and two-dwelling building in Sweden is a building built in 1953. As previously mentioned, this is likely to be a mean value; an analysis based on median values would show that the most common one- and two-dwelling building was built during the million homes programme.

The most detailed study from Boverket [32] based its analysis on two reference buildings: a 1.5-storey building and a two-storey building. Both reference buildings have wooden constructions for the exterior walls and roof, and also the façades are wooden. None of the reference buildings include a cellar. By including both a 1.5-storey building and a two-storey building, the study included almost all existing roof constructions. However, by only including wooden façades, roughly 50% of the façade constructions were excluded. Cellars, which may be found in roughly 30% of the existing one- and two-dwelling buildings, were not included in the study.

A recent study that included renovation measures [21] included two different types of exterior walls: wood and lightweight concrete. Further studies on one- and two-dwelling buildings could exclude exterior walls with lightweight concrete and increase their focus on different types of roof constructions and/or façade materials, as they vary more.

The study that suggested 46 different typical buildings to represent the existing building stock [46] argued that a 1.5-storey building was the most common building type to be built during the 1970s and should be the choice of a typical building as it would represent 65% of the buildings from that decade. However, looking at the available data, the most common building has one storey. Roughly 40% of the buildings from the 1970s have one storey.

## 7. Conclusions

This study shows the importance of studying differences among building constructions in the existing building stock when studying renovation measures and analysing the renovation potential. It

shows that there is a set of constructions and building techniques that were commonly used in the existing building stock in Sweden during the million homes programme and in the decades before and after. It is important to underline that there is no single construction type or building that has been predominant. Furthermore, regional differences exist.

Previous studies have often assumed a rather large homogeneity and did not always include the most common constructions and building types in their studies. This concerns studies regarding the development of prefabricated building elements and studies regarding cost calculations for renovation and energy efficiency measures. Furthermore, studies that used a large set of buildings to create a building typology created typologies that cannot be confirmed by the data in this study.

If assumptions of large homogeneity are misjudged, they may cause higher costs for renovation measures than predicted, and developed prefabricated building elements may apply on fewer buildings than expected. This may limit the reliability of potential studies and slow down the renovation pace or limit the actual renovation measures.

To speed-up the renovation of the existing building stock in the EU and in other regions, further studies are needed to form a basis for making well-informed decisions regarding political directives and incentives and regarding actual renovation measures. Furthermore, as buildings will always be unique, the development of prefabricated building systems needs to have flexibility to enable their use on a larger scale.

Based on the available data, it is possible to draw some conclusions regarding construction types, which should be prioritised in further research regarding the Swedish building stock.

### 7.1. Multi-Dwelling Buildings

The most commonly used façade materials in multi-dwelling buildings are clay bricks, render and concrete. Façades with clay bricks are common throughout Sweden. The most common inner material used for clay brick façades is wood, which indicates that it is most likely found in light infill walls. Rendered façades are most common in the Stockholm region and are also rather common in non-metropolitan regions. The rendered façades are almost exclusively paired with lightweight concrete as the inner material. Concrete façades are common in Malmö, Göteborg and Stockholm regions. Concrete façades are almost exclusively paired with concrete as the inner material. Based on these findings, the most common constructions that should be investigated in future studies are summarised in Table 1. The constructions include both load-bearing walls and light infill walls in the existing building stock.

**Table 1.** Summary of common constructions for exterior walls in multi-dwelling buildings in Sweden produced in the post-war period.

| Type of Construction | Region | | | |
|---|---|---|---|---|
| | Non-Metropolitan | Malmö-Region | Göteborg-Region | Stockholm-Region |
| Insulated wood infill walls with clay brick façades | X | X | X | X |
| Lightweight concrete walls with rendered façades | X | | | X |
| Concrete sandwich walls | | X | X | X |

### 7.2. One- and Two-Dwelling Buildings

The most common façade materials used in one- and two-dwelling buildings are clay bricks and wood. Together, these two materials represent more than 80% of the dwellings from the studied period. Almost all exterior walls are constructed with wood as the inner material. Furthermore, roof constructions with an insulated tie beam and roof constructions where the tie beam is also part of an interior floor slab (in 1.5-storey buildings) need to be studied. Based on these findings, the most

common constructions for further studies are summarised in Table 2. As can be seen, there are no regional differences regarding the most common constructions.

**Table 2.** Summary of common constructions for exterior walls and roofs in one- and two-dwelling buildings in Sweden constructed in the post-war period.

| Type of Construction | Region | | | |
|---|---|---|---|---|
| | Non-Metropolitan | Malmö-Region | Göteborg-Region | Stockholm-Region |
| Insulated wood walls with clay brick façades | X | X | X | X |
| Insulated wood walls with wood façades | X | X | X | X |
| Roof constructions with insulated tie beam | X | X | X | X |
| Roof constructions for 1.5-storey buildings | X | X | X | X |

**Supplementary Materials:** The following are available online at http://www.mdpi.com/2075-5309/9/4/99/s1.

**Author Contributions:** data curation, B.B.; visualization, B.B.; writing—original draft, reviewing and editing, B.B. and M.W.

**Funding:** This study was funded by The Development Fund of the Swedish Construction Industry (SBUF) and Skanska Sverige AB as part of the project; "Klimatskal 2019", which aims to develop robust renovation measures for existing building envelopes in Sweden.

**Conflicts of Interest:** The authors declare no conflict of interest.

## Appendix A. —Multi-Dwelling Buildings

**Table A1.** Available data for multi-dwelling buildings, via supplementary data file.

| Type of Data | Available Parameters | Period for Available Data |
|---|---|---|
| Type of building | Balcony access building; Point block building; Slab block building; Terraced building; Other | 1960–1993 * |
| Area of construction | Göteborg region; Malmö region; Stockholm region; Sweden excluding metropolitan regions | 1966–1993 ** |
| Storeys | 1–2; 3; 4; 5–8; ≥9 | 1962–1993 |
| Type of superstructure | Transverse load bearing; Longitudinal load bearing; Pillar construction; Other | 1968–1972 |
| Material for superstructure | Autoclaved aerated concrete; Clay bricks; Concrete; Wood; Other | 1963–1987 |
| Method of production for superstructure | On site; Prefabricated | 1968–1979 |
| Façade material | Asbestos; Autoclaved aerated concrete; Clay bricks; Concrete; Render; Sandlime bricks; Sheet metal; Wood; Other | 1963–1993 *** |
| Inner material in exterior wall | Autoclaved aerated concrete; Clay bricks; Concrete; Wood; Other | 1963–1979 |
| Method of production for exterior wall | On site; Prefabricated | 1968–1979 |
| Roofing | Asbestos; Clay tiles; Concrete tiles; Roof felt; Sheet metal; Other | 1969–1993 |

* Terrace buildings were reported separately from 1979. Before 1979; they are included in "Other"; ** From 1987, data regarding façade material were reported by western Sweden (expanded Göteborg region), southern Sweden (expanded Malmö region) and eastern Sweden (expanded Stockholm region); *** Up until 1980, the main material for façades was reported if a mix of different façade materials were used. 1980–1993, mixed façades were reported including the two main materials.

## Appendix B. —One- and Two-Dwelling Buildings

**Table A2.** Available data for one- and two-dwelling buildings, via supplementary data file.

| Type of Data | Available Parameters | Period for Available Data |
|---|---|---|
| Type of building | One-dwelling building; Two-dwelling building; Linked building; Terraced building; Other | 1966–1994 |
| Area of construction | Göteborg region; Malmö region; Stockholm region; Sweden excluding metropolitan regions | 1968–1994 |
| Storeys | 1; 1.5; 2; >2 | 1970–1987 |
| Material for superstructure | Autoclaved aerated concrete; Clay bricks; Concrete; Wood; Other | 1966-1987 |
| Method of production for superstructure | On site: Prefabricated; Partly prefabricated | 1968–1993 |
| Façade material | Asbestos; Autoclaved aerated concrete; Clay bricks; Concrete; Render; Sandlime bricks; Sheet metal; Wood; Other | 1966–1993 * |
| Insulation in exterior wall | Expanded polystyrene; Wood insulation/wood wool/wood shavings; Autoclaved aerated concrete; Mineral wool | 1966–1972 |
| Roofing | Asbestos; Clay tiles; Concrete tiles; Roof felt; Sheet metal; Other | 1966–1993 |

* Up until 1973, the main material for façades was reported if a mix of different façade materials was used. 1973–1993, mixed façades were reported including the two main materials.

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
