# Peer review of "Review of Constructions and Materials Used in Swedish Residential Buildings during the Post-War Peak of Production"

_buildings, doi:10.3390/buildings9040099_

Reviewer 1 Report

The paper presents the assessment of energy performances in the existing building stock, in order to increase the renovation pace. Departing from a more large framework, the study focuses on the Swedish  residential building stock. 

The paper is really interesting and very well documented. The introduction discussed the main topics related to the research, considering a legislative framework and procedural methodologies. 

To have a general discussion on the evaluation of energy performance of the building stock, comparing different methodologies, I suggest the paper: Nardi ed al., Quantification of heat energy losses through the building envelope, Building and Environment, 146,  2018, 190-205. Here you can find a support to understand the importance of the building envelope in the energy balance of buildings, comparing different traditional procedures (also the one related to Tabula you cited). 

The section definition and nomenclature mus be inserted in an appendix. Here you can describe better the different typologies normally used in Sweden, also inserting a map for their location. 

The structure of the paper is not clear. Please, add a section on the methodology.  Also, you can organize the paper dividing the methodology and the results. In this way, you will the comprehension of the results. 

You n the part material and methods, it is not clear which energy diagnosis procedure you use. Do you use non distruttive technologies? Infrared thermography ? How do you recognize water problems? Do you consider also the influence of the percentage of humidity in the thermal performance calculation? In this case, I suggest to read: Lucchi, Non-invasive method for investigating energy and environmental performances in existing buildings, PLEA 2011 - Architecture and Sustainable Development, Conference Proceedings of the 27th International Conference on Passive and Low Energy Architecture, 2011.

Conclusion are very concise, but contains the most interesting results of the paper. 

Author Response

Thank you for your review and your comments. Much appreciated.

I have considered your remarks, revising the manuscript.

Furthermore, English language editing will purchased/carried out via MDPI language editing after your review of the revised manuscript.

Below you´ll find response/comments to your review.

Regarding suggested paper (Nardi et al 2018).

The paper presents a comprehensive review of approaches for U-value evaluation, including IRT. Reference to the paper is included in revised version.

Regarding moving the section “definition and nomenclature”.

As the other reviewers did not give this suggestion. We will ask the editor of Buildings/MDPI to make this decision since it is related to the format of the journal. If they say it should be moved, we will move it.

Regarding suggestion of a adding a methodology section.

A methodology section has been added.

Regarding question related to energy diagnosis procedure.

No energy diagnosis is made. Hence, there is no procedure to present.

Reviewer 2 Report

1.       Authors have not presented any   material about how their analysed buildings are renovated at the moment and   what they state now. Could this information be presented? Moreover, it is   helpful to have this date for further restoration of buildings to meet nowadays   standards and needs of habitants. It is hard to believe that those dwellings   are not renovated (can be partially) till now. Some considerations can be presented   in discussion.

2.       There is luck of method authors   used to evaluate or to assess dwellings. Can authors describe their used   assessment method in more detail?

3.       What is the current state of the   analysed dwellings? Have they been renovated somehow? Because, evaluating a   current state of buildings, but not only documents, is important in future   restoration of dwellings to meet nowadays standards and needs of habitants.

4.       Have authors evaluated the meet of   their evaluated building to standards and norms used after 2018 till now in   their paper?

5.       A part of analysed literature   should be renewed. There are a number of newer researches than authors   present. At the moment there are 17 among 50 papers after 2015 year.

Author Response

Thank you for your review and your comments. Much appreciated.

I have considered your remarks, revising the manuscript.

Furthermore, English language editing will purchased/carried out via MDPI language editing after your review of the revised manuscript.

Below you´ll find response/comments to your review.

Regarding question 1, 3 and 4, related to how analysed buildings are renovated at the moment.

Nu such data is available in Sweden. However, some general information and estimations may be found in reports from the Swedish National Board of Housing, Building and Planning (Boverket). The reports are mentioned in the introduction of the manuscript. We have revised the manuscript and included the available information from the reports.  

Regarding question 2, related to method.

A methodology section has been added.

Regarding question 5, related to need for more recent research.

More recent research has been added.

Reviewer 3 Report

See the attached document, which is the review for the Authors.

Author Response

Thank you for your review and your comments. Much appreciated.

I have considered your remarks, revising the manuscript.

Furthermore, English language editing will purchased/carried out via MDPI language editing after your review of the revised manuscript.

Below you´ll find response/comments to your review.

Regarding comment: “abstract seems to be the introduction”.

The abstract included in the original version was not an introduction. Revised abstract in the manuscript had been changed to further highlight purpose, scope, procedure/methodology and major findings.

Regarding comment “the introduction does not provide sufficient background information to understand the problem, the study, and the results.

Please see line 58-71 in orig. submitted manuscript (line 61-74 in revised version) for description of problem.

Please see line 110-114 in orig. submitted manuscript (line 122-126 in revised version) for description of the study and what the results may be.

Also see line end of introduction line 115-120 in orig. submitted manuscript (line 127-132 in revised version) which gives an overview of the paper.

Regarding suggestion to add more references.

More references are added.

Regarding suggestions to change pillar to columns and facade to façade.

This is changed in revised version. Furthermore, English language editing will purchased/carried out via MDPI language editing after your review of the revised manuscript.  

Regarding comment on section 3.

A methodology section has been added.

Round  2

Reviewer 1 Report

-

Author Response

Thank you for your review and your comments. Much appreciated.

Reviewer 2 Report

Accept in present form

Author Response

(The authors gave the same response as above.)

Reviewer 3 Report

See the attached document, which is the review for the Authors.

Author Response

Thank you for your review and your comments. Much appreciated.

Below you´ll find response/comments to your review.

 Regarding the abstract:

The abstract includes all parts of the article; introduction, materials and method, results and conclusions. It should be a very short summary of the paper, including all parts. Thus, also the introduction (challenge and need of the study) is included.

We have made minor changes to further highlight the scope and procedure.

 Regarding suggestions to cite P. Foraboschi. Structural layout that takes full advantage of the capabilities and opportunities afforded by two-way RC floors, coupled with the selection of the best technique, to avoid serviceability failures. Engineering Failure Analysis,

2016; 70(December): 387-418:

The suggested article presents an interesting case study about the redesign of a floor construction including designing and constructing a test building. However, the article does not deal with description and mapping of an existing building stock (which is the main purpose of our study), therefore it would be confusing to add this article.

 Regarding suggestions to cite P. Foraboschi. Versatility of steel in correcting construction deficiencies and in seismic retrofitting of RC buildings. Journal of Building Engineering, 2016; 8(December): 107-122:

The suggested article is an interesting article, presenting the structural upgrade of a public school building in Montelabbate, saving the school from demolition, maintaining the building's architectural integrity. However, the article does not deal with a description of an existing (residential) building stock (which is the main purpose of our study), therefore it would be confusing to ad this article.

 Regarding comment on section 3, not being well explained and that it is not possible for someone else to repeat the study:

The material that the study is based on is described in Chapter 3 and in Appendix A and B. The method is described in Chapter 4. All the compiled data is available for other researchers for further studies. If further details are asked for in order to make additional research based on the data, researchers are welcome to contact the author.

 Regarding comment on the need for including windows and glass envelopes and suggestion to cite two additional articles from P Foraboschi (2014 & 2017):

Information about window types were not included in the database and can therefore not be added in the article. The study is limited to opaque building envelopes. Largely glazed envelopes were not usual in residential buildings during the period 1962-1992 in Sweden. Therefore, the additional two articles by Foraboschi, focusing on glass, are not used as references.

 Regarding the comment related to the need to better explain differences between transverse and longitudinal load-bearing and information related to how the slabs are made:

We describe the available data, more detailed data does not exist and cannot be presented.

Round  3

Reviewer 3 Report

The article has no scientific structure. 

That was the main flaw of the original submission, which the Authors did not eliminate in the following resubmissions.

Thus, the revised version resubmitted does not deserve publication in a scientific archival journal.